# Unmasking the Deceptive Nature of Cancer Stem Cells: The Role of CD133 in Revealing Their Secrets

**DOI:** 10.3390/ijms241310910

**Published:** 2023-06-30

**Authors:** Julia Pospieszna, Hanna Dams-Kozlowska, Wachirawit Udomsak, Marek Murias, Malgorzata Kucinska

**Affiliations:** 1Department of Toxicology, Poznan University of Medical Sciences, 30 Dojazd Street, 10 Uniwersytetu Poznanskiego Street, 60-631 Poznan, Poland; julia.pospieszna@gmail.com (J.P.); wachirawit5414@gmail.com (W.U.); kucinska@ump.edu.pl (M.K.); 2Department of Cancer Immunology, Poznan University of Medical Sciences, 15 Garbary Street, 61-866 Poznan, Poland; 3Department of Diagnostics and Cancer Immunology, Greater Poland Cancer Centre, 15 Garbary Street, 61-866 Poznan, Poland; 4Center for Advanced Technology, Adam Mickiewicz University in Poznan, Uniwersytetu Poznanskiego 10 Street, 61-614 Poznan, Poland

**Keywords:** CD133, prominin-1, cancer stem cells, gene therapy, immunotherapy, nanotechnology

## Abstract

Cancer remains a leading cause of death globally, and its complexity poses a significant challenge to effective treatment. Cancer stem cells and their markers have become key players in tumor growth and progression. CD133, a marker in various cancer types, is an active research area as a potential therapeutic target. This article explores the role of CD133 in cancer treatment, beginning with an overview of cancer statistics and an explanation of cancer stem cells and their markers. The rise of CD133 is discussed, including its structure, functions, and occurrence in different cancer types. Furthermore, the article covers CD133 as a therapeutic target, focusing on gene therapy, immunotherapy, and approaches to affect CD133 expression. Nanoparticles such as gold nanoparticles and nanoliposomes are also discussed in the context of CD133-targeted therapy. In conclusion, CD133 is a promising therapeutic target for cancer treatment. As research in this area progresses, it is hoped that CD133-targeted therapies will offer new and effective treatment options for cancer patients in the future.

## 1. Introduction

Based on the data published by the International Agency for Research on Cancer (IARC), cancer is a significant global health issue, with nearly 10 million cancer-related deaths and 19.3 million new cases diagnosed in 2020. The most frequently diagnosed cancers include the breast, lung, and colorectal cancers [1]. According to the World Health Organization (WHO), the number of newly diagnosed cancer cases is expected to increase by 47.4% in 2040 compared to 2020 [1]. Therefore, cancer prevention, diagnosis, and treatment require continuous attention and improvement [2].

Despite a better understanding of the molecular mechanism involved in all cancer stages, its treatment is still a huge challenge. One reason is tumor heterogeneity and the presence of cell populations with different genetic and phenotypic properties. Tumors may exhibit different characteristics depending on the type of cells involved, such as growth rate, the ability to invade surrounding tissues, and sensitivity to treatment; the situation becomes even more complicated when we consider the existence of subpopulations of cancerous stem cells [3]. Therefore, understanding the role of different subpopulations of cancer cells in the development and progression of cancer is an active area of research that could lead to more effective cancer treatments in the future [4]. 

The initiation of tumors can occur due to mutations caused by various factors, including infections, toxins, radiation, or metabolic influences in transformed differentiated cells or tissue-resident stem cells [5]. It may be mediated by protooncogenes which control various biological processes in normal cells, serving as growth factors, transducers of cellular signals, and nuclear transcription factors [6]. In mammalian genomes, protooncogenes control normal cell differentiation and proliferation [7]. Any alterations to these genes that influence the structure of their encoded proteins may result in the development of oncogenes in cancer cells. The oncogenes then act as a driver of cell division, significantly affecting cancer development [8]. During transformation, oncogenes are overexpressed, and tumor suppressors are deactivated, leading to the uncontrolled growth of cells. Differentiated cells require more genomic changes for transformation than tissue-resident stem cells, which require only a few genomic changes [9]. Cancer stem cells (CSCs) are a small subset of cancer cells responsible for tumor initiation, maintenance, metastasis, or relapse. CSCs are a small subset of cancer cells responsible for tumor initiation, maintenance, metastasis, or relapse. The origin of CSCs is still not fully understood, but two main theories exist: the stochastic and hierarchical models (Figure 1). CSCs are believed to arise from a specific subset of cells that possess stem cell properties and are derived from normal tissue stem cells or progenitor cells that have acquired additional mutations [5].

Like stem cells of non-pathological tissues, CSCs possess self-renewal properties, allowing them to maintain the CSC pool and contribute to tumor growth [9]. Cancer stem cells possess the property of “robustness”, which refers to several biological characteristics mediated by numerous mechanisms [10]. The most important feature is multidrug resistance (MDR) manifested by an enhanced ability to expel anticancer drugs through ATP-binding cassette transporters as well as the capacity to carry out the rapid repair of damaged DNA, the ability to adapt to a hyperinflammatory or hyponutritious microenvironment, plasticity in the transition to transit-amplifying cells, and metabolic reprogramming [10,11,12]. Such characteristics of CSCs are responsible for the formation of MDR, corresponding to clinically undetectable lesions enriched in CSCs that remain after therapy and later give rise to cancer progression as a relapse or distant metastasis [10]. These mechanisms allow CSCs to survive and contribute to tumor recurrence after treatment (Figure 2) [13,14,15,16]. In summary, cancer stem cells are crucial for cancer treatment, and understanding their properties could lead to the development of a more effective therapeutic approach for cancer in the future [17,18]. 

Hence, targeting the cancer stem cell population for effective cancer treatment is imperative. Targeted therapy is a treatment modality that focuses on specific molecules involved in the growth and proliferation of cancer cells, intending to eliminate cancer cells while minimizing damage to non-cancerous cells [19]. Various studies have demonstrated that targeted therapies targeting CSCs can effectively impede their ability to regenerate and repopulate the tumor, leading to tumor size reduction and even complete remission [5]. However, it is crucial to acknowledge that targeted therapies are not a definitive cure for cancer, and further research is warranted to comprehend their full potential as a treatment option. Therefore, CSC-targeted therapy might be a critical adjunct to conventional anticancer treatment approaches. To augment the effectiveness of CSC-targeted therapy, surface markers can be utilized. Many cancer stem cell markers can be employed in CSC-targeted therapy, with some identified as correlated with CSCs in diverse cancer types (Table 1). 

Interestingly, CSCs and embryonic stem cells share similar traits, particularly in their developmental signaling pathways, which control the self-renewal of stem cells [69]. The activation of these highly conserved pathways can, at least in part, be responsible for their resistance [70]. CSCs rely on critical pathways such as Hedgehog, Wingless-related integration site (Wnt/β-catenin), Notch, Janus kinase/signal transducer and activator of transcription (JAK/STAT) and nuclear factor erythroid 2-related factor 2 (Nrf-2) and it is commonly accepted that their dysregulation leads to differences in treatment resistance, metastasis, and proliferation between cancerous and normal stem cells [71,72,73,74]. For example, in female-specific [75] and hepatic cell carcinoma [37], deregulated Notch signaling can stimulate self-renewal in CSCs. Additionally, the interconnectivity of signaling pathways can impact downstream effectors [69,76]. CSCs also activate other signaling pathways like transforming growth factor β1 (TGF-β) [77], phosphatidylinositol-3-kinase and protein kinase B (PI3K/Akt) [78,79], and epidermal growth factor receptor (EGFR) [18,80], and transcriptional regulators, such as the sex determining region Y-box (SOX-2) [81,82], cellular myelocytomatosis oncogene (c-Myc) [81,83], NANOG [81,84], and octamer-binding transcription factor 4 (OCT4) [81,85] to maintain their self-renewal and differentiation capabilities. Moreover several reports have demonstrated the significant contribution of Wnt [86,87], Notch [88,89], Hedgehog [90,91,92], and Hippo [93,94] signaling pathways to metastasis. Importantly, it is crucial to recognize that the surface markers associated with CSCs can vary depending on the type of cancer and not all CSCs may express the same surface markers (Table 1). Consequently, targeted therapy should be based on comprehensive research on the patient’s characteristics, encompassing the paradigm of personalized medicine. Continuing research is underway to better understand CSCs’ surface markers and how they can be leveraged to identify and target these cells for cancer treatment. One such extensively studied biomarker used for CSC isolation is CD133 [13,14].

The fundamental trends in the co-occurrence of key terms “cancer stem cells” and “CD133” with other key terms in the papers published in the last 10 years are shown in Figure 3. 

The above figure shows how much the CD133 protein has been studied in different aspects. These studies aim to better understand the properties and biological functions concerning cancer as a disease. All this research gives a huge step to transferring the knowledge to practical application and might open new ways for cancer treatment and overcoming the resistance to currently used therapies. In this review, we briefly described the functions of CD133 and its role in physiological processes and the “dark side” in cancer development and resistance to treatment. We discussed the possibilities of using CD133 as a molecular target and focused on gene therapy, immunotherapy, chemotherapy, and photodynamic therapy. These treatment options were selected as the most studied in recent years and seemed more promising to improve therapeutic effectiveness. Finally, we focused on the CD133 as a drug delivery system based on nanocarriers for different cargoes to target cancer stem cells selectively.

## 2. A Brief 26-Year History of CD133: From Discovery to Understanding the Role of Protein 

### 2.1. The Discovery of CD133

In 1997, two independent groups described a novel pentaspan membrane glycoprotein [95,96,97]. The first group of researchers discovered a glycoprotein called prominin the mouse neuroepithelium [95]. The other group examined a molecule called AC133 in acute myeloid leukemia [96,97]. Both prominin-1 and AC133 were later found to be the same protein, now known as CD133. 

For over 20 years, scientists have proposed various roles for prominin-1, including its function as a stem cell and cancer stem cell biomarker, its involvement in organizing plasma–membrane protrusions, maintaining the apical-basal polarity of epithelial cells, biogenesis of the photoreceptive disc, mechanism of multidrug resistance, and the capacity for self-renewal and tumor formation [98]. 

Prominin-1 was firstly discovered in hematopoietic stem and progenitor cells [96,97]. However, nowadays there is a lot of evidence showing that CD133 is considered a biomarker for several types of cancer, including ovarian cancer, colorectal cancer, lung cancer, breast cancer, hepatocellular cancer or gastric cancer [99,100,101,102,103,104,105,106,107,108,109] (Table 1).

### 2.2. The Receptor Structure

CD133 is a glycoprotein which contains five transmembrane domains. It also has two extracellular loops and two smaller intracellular loops. Human prominin-1 is coded by PROM1 gene located on chromosome 4 (4p15.32) and contains 34 exons [110]. CD133 is 865 aminoacids long and has 97 kDa and around 130 for unmodified and glycosylated protein, respectively [111]. Figure 4 shows the structure of the CD133 glycoprotein.

### 2.3. CD133 Physiological Functions—Cell Differentiation, Proliferation, and Survival

CD133 has been shown to play a critical role in maintaining of stemness and self-renewal capacity in embryonic and neural stem cells [112,113]. CD133 expression has been observed in several tissues during embryonic growth, including the developing nervous system, retina, kidney, and liver [114]. In retinal development, CD133 is first expressed in the developing optic vesicle and is maintained in retinal progenitor cells throughout retinal development.CD133-positive retinal progenitor cells that have been isolated from human fetal retinas have the potential to differentiate into multiple retinal cell types, while CD133 has been shown to play a critical role in maintaining retinal progenitor cell proliferation and differentiation [115,116]. CD133-positive neural stem cells in the nervous system can differentiate into various cell types, including neurons, astrocytes, and oligodendrocytes. CD133 has also been shown to play a role in preserving the self-renewal capacity of neural stem cells and promoting their migration to appropriate locations during development [117,118,119]. CD133 expression has also been observed in the adult brain, particularly in regions of active neurogenesis, such as the subventricular zone and the dentate gyrus of the hippocampus [118,120]. CD133-positive NSCs (neural stem cells) in these regions can differentiate into new neurons, suggesting a potential role for CD133 in maintaining adult neurogenesis and neuronal plasticity [118].

CD133-positive RPCs (renal progenitor cells) can differentiate in the kidney into various cell types, including proximal and distal tubular epithelial cells, podocytes, and interstitial cells. CD133 has also been suggested to regulate cell proliferation and differentiation in the developing kidney. CD133 expression has been observed in the adult kidney, particularly in active repair and regeneration regions, such as the proximal tubules following injury. CD133-positive cells in these regions can differentiate into new tubular epithelial cells, suggesting a potential role for CD133 in the maintenance of adult renal repair and regeneration [121,122]. In the liver, CD133 has been identified as a marker for hepatic progenitor cells [123]. CD133-positive hepatic progenitor cells can differentiate into various cell types in the liver, including hepatocytes, cholangiocytes, and stellate cells [123,124,125]. CD133 has also been suggested to regulate cell proliferation and differentiation in the developing liver. CD133 expression has also been observed in the adult liver, particularly in active repair and regeneration regions, such as the periportal area following injury [126,127]. CD133-positive cells in these regions can differentiate into new hepatocytes and bile duct cells, suggesting a potential role for CD133 in maintaining of adult liver repair and regeneration [128]. Moreover, CD133 expression has been observed in photoreceptor and bone marrow cells [124,125,129]. CD133 is required to form the outer segment discs in rod photoreceptor cells [130]. CD133-deficient mice have been shown to have abnormal outer segments and impaired visual function. In bone marrow cells, CD133 expression has been observed in a subpopulation of hematopoietic stem and progenitor cells, which have a greater colony-forming capacity and multi-lineage differentiation potential compared to CD133-negative cells. CD133 has also been implicated in regulating cell proliferation, survival, and differentiation in these cells [131].

In addition to its role in maintaining stemness and a self-renewal capacity in various stem cell populations, CD133 has been found to interact with several molecular pathways and signaling networks that regulate CSCs. One such pathway is the PI3K/Akt/mTOR pathway, which is crucial for cell growth, survival, and metabolism [132,133] CD133 has been shown to activate this pathway, promoting CSC proliferation and survival. This interaction also contributes to the resistance of CSCs to chemotherapy and targeted therapies [134]. Furthermore, CD133 has been implicated in the regulation of the epithelial–mesenchymal transition (EMT), a process associated with increased CSC properties, tumor invasion, and metastasis. CD133 expression has been linked to the induction of EMT-related transcription factors, such as Snail, Twist, and ZEB1, promoting CSC invasiveness and metastatic potential [135]. Additionally, CD133 has been shown to modulate the activity of the NF-κB pathway, a central regulator of inflammation and tumor progression [136,137]. CD133 expression enhances NF-κB activation, leading to increased CSC survival, resistance to apoptosis, and promotion of an inflammatory microenvironment that supports CSC maintenance [136]. Moreover, CD133 has been reported to interact with the TGF-β signaling pathway, which plays a pivotal role in CSC plasticity and immune evasion. CD133 expression contributes to TGF-β-induced CSC phenotypic changes and immunosuppressive effects, promoting tumor growth and immune resistance [138]. Collectively, these interactions highlight the intricate involvement of CD133 in multiple molecular pathways and signaling networks that regulate CSC properties and tumor progression, offering potential therapeutic targets for disrupting CSC-mediated tumor growth and therapy resistance.

In summary, the expression of CD133 in embryonic and adult stem cells in various tissues highlights its potential significance in developing regenerative medicine strategies to promote tissue repair and regeneration in neurodegenerative diseases, visual impairments, kidney disease, liver disease, or blood disorders.

### 2.4. The Role of CD133 Glycosylation

Glycans are present in the form of attachments to proteins, such as in the cases of glycoproteins and proteoglycans. Typically, glycans are located on the external surface of cells and perform a vital function in numerous physiological processes. Dysregulated glycosylation is crucial in the tumors’ initiation, promotion, progression, and aggressiveness [139]. Moreover, most CSC markers are glycoproteins, including CD44, CD24, and CD133, that express various glycan moieties on their cell surface [140,141]. Glycosylation of CSC markers modulates several CSC functions, including cell adhesion, immune cell evasion, extravasation, apoptosis, self-regeneration, and pluripotency preservation [139]. Furthermore, glycosylation of CD133 has been suggested as a secondary CSC indicator [113] In certain glioma CSCs, sialylation of CD133 N-glycan terminal via the α2,3-site is augmented in hypoxic conditions, correlating with brain CSC migration and survival [140,142]. Moreover, removing sialic acid from CD133 through neuraminidase results in a specific enhancement of its degradation via a pathway that depends on lysosomes [143]. The co-expression of CD133 and the sialyl-Thomsen-nouveau antigen (STn), a carbohydrate moiety found on protein markers of CSCs in ovarian cells and which has been connected with the inability to eliminate quiescent cancer stem cells that survive chemotherapy, is believed to be a contributing factor to recurrent ovarian cancer and tumor recurrence [144]. Therefore, treatment methods targeting abnormal glycosylation seem promising options for effective tumor therapy. Inhibiting glycans can reduce CSCs’ ability to maintain stemness, thus reducing tumor proliferation. Because STn is co-expressed with CD133, anti-STn antibody–drug conjugates can suppress tumor growth and effectively reduce CSCs [142]. Other antibodies, such as the anti-sialyl-di-Lewis antibody FG129, are being tested to target tumor-associated glycans and develop tumor-selective treatments and diagnostic modalities [145]. Additional research is needed to distinguish the glycome and glycosylation differences among normal cells, stem cells, CSCs, and non-CSCs, despite the recognition of glycosylation’s crucial involvement in CSC signaling pathways and markers regulating self-renewal, stemness, and extravasation. This information may enable the development of biomarkers for detecting cancer progression and allow researchers to accurately target cancer cells and resistant CSCs. Increasing evidence shows that the abnormal glycosylation of CSCs plays a critical role in their ability to resist chemotherapy and metastasize through several pathways. Although inhibiting or manipulating glycosylation in CSCs has demonstrated therapeutic potential, further exploration of the associated glycosylation processes is necessary to devise effective strategies targeting specific altered markers or signaling pathways without affecting healthy cells [142]. Approaches such as selectively cleaving the surface glycan of tumors or drugs with an affinity for tumor-associated glycans have demonstrated varying toxicity to cancer and normal cells, indicating potential therapeutic window optimization [146,147]. Although CSC markers may exhibit intratumoral and intertumoral heterogeneities, glycosylation may provide relevant targets, such as sialyl acid, preserved throughout tumors, simplifying the development of effective and extensive treatment strategies [142].

## 3. CD133 as a Target in Cancer Therapies

CD133-expressing CSCs have been shown to exhibit resistance to chemotherapy and radiation therapy, and are associated with poor prognosis in various cancers [105]. Thus, targeting CD133 overexpressing CSCs has emerged as a potential rational therapeutic strategy for improving cancer treatment outcomes. To date, by targeting CD133 in cancer cells, different cargoes can be delivered specifically to the cancer stem cell population. On the other hand, affecting the CD133 gene expression using the small interfering RNA (siRNA)/short hairpin RNA (shRNA) or chemotherapeutics can reduce the stemness potential of cancer cells. 

Various approaches have been explored for targeting CD133 on CSCs, including (i) gene therapy that uses CD133-targeted vectors to deliver therapeutic genes to CSCs, (ii) monoclonal antibodies (mAb) that specifically recognize and bind to CD133 and other immunological strategies, (iii) nanoparticles or nanoliposomes which can deliver therapeutic agents to the cells expressing CD133 selectively, and (iv) oligonucleotide therapeutics or chemotherapeutics which, by decreasing the level of CD133, reduce the cancer cell stemness property. These strategies have shown promising preclinical results and are being evaluated in clinical trials. On the other hand, the heterogeneity of CSCs and the potential for off-target effects pose challenges to developing effective CD133-targeted therapies.

### 3.1. Strategy for Gene Therapy

Gene therapy is a promising approach for treating genetic and acquired diseases by delivering genetic material to specific cells. Strategies for targeted gene transfer have been developed to precisely deliver therapeutic nucleic acids to specific cells, tissues, or organs [148]. Genetic material can promote the expression of the transferred gene, suppress the expression of a target gene, or modify a target gene.

Anlinker and coworkers [149] reported a targeted gene transfer method using a lentiviral vector that can specifically target neurons, endothelial cells, and hematopoietic progenitors (HPCs) by fusing the coding sequence of a single-chain fragment variable (scFv) for CD133 to the reading frame of a cytoplasmic tail-transduced mutant or wild-type measles virus hemagglutinin. This approach has been used to selectively deliver genes to CD133-expressing human glioblastoma stem cells (GSCs) [150]. This vector selectively transduced only CD133^+^ GSCs but not CD133^−^ cells or normal brain cells. In addition, the CD133-specific promoter was incorporated into the viral vector to control transgene expression and was found to be more effective in GSCs than other promoters.

Moreover, shRNA was used to inhibit the expression of the CD133 gene in glioblastoma stem cells that expressed CD133. shRNA is a short RNA fragment capable of blocking or inhibiting the expression of specific genes in cells. The study suggests that the CD133-specific lentiviral vector system could be used for selective gene delivery and targeted therapy for glioblastoma, a highly aggressive brain cancer with a poor prognosis [100,149,150]. This study showed that CD133 could not only be a tool to target CSCs, but that downregulation of its expression could be beneficial in cancer therapy. Furthermore, a study investigated the combined effect of CD133 siRNA and Oxaliplatin on the proliferation, migration, apoptosis, and stemness properties of colorectal cancer cells (HT-29 cell line) [151]. The results showed that the combination therapy reduced the IC_50_ value of Oxaliplatin, suppressed the CD133 mRNA and protein level, reduced the migration and stemness properties of the cells, and stimulated cell death. Therefore, the knockdown of CD133 could be a promising strategy to sensitize colorectal cancer cells to Oxaliplatin and improve the effectiveness of treatment. However, further research is needed to address limitations such as the recycling of CD133 after transient silencing [151].

### 3.2. The Relationship between CD133 and Chemotherapeutic Drugs 

Cancer chemotherapy is a widely used treatment modality in the management of cancer. It involves the administration of specific drugs that have the ability to kill or inhibit the growth of cancer cells. Chemotherapy aims to target cancer cells throughout the body, including those that may have spread from the primary tumor to other sites, to achieve a therapeutic response and improve patient outcomes [152]. Several studies have shown that directly targeting CD133 with monoclonal antibodies, aptamer, antibody fragments, and other advanced delivery systems might transport chemotherapeutic drugs effectively to cancer stem cells [153,154,155,156,157]. 

The study conducted by Zhou et al. investigated the effects of CD133 expression on chemotherapy response and drug sensitivity in adenoid cystic carcinoma (ACC) [158]. The study aimed to explore the role of CD133 in determining the efficacy of chemotherapy in ACC. CD133 expression levels were analyzed in ACC tumor samples and correlated with the response to chemotherapy and drug sensitivity. It was found that high CD133 expression was significantly associated with a reduced response to chemotherapy and increased resistance to drugs commonly used in ACC treatment [158]. The study highlights the potential significance of CD133 as a predictive biomarker for chemotherapy response in ACC. The findings suggest that targeting CD133-positive cancer stem cells may improve the effectiveness of chemotherapy and overcome drug resistance in ACC patients

The other study aimed to investigate the impact of targeting CD133 on the chemotherapeutic efficacy of recurrent pediatric pilocytic astrocytoma (PA) following prolonged chemotherapy [159]. The researchers sought to determine whether CD133 could be a potential therapeutic target to enhance treatment outcomes in this brain tumor. The study utilized in vitro models to evaluate the effects of targeting CD133 in recurrent PA. Firstly, the presence of CD133-positive cells in patient-derived PA samples was confirmed. They then examined the efficacy of CD133-targeted chemotherapy (doxorubicin, vinblastine, vincristine) in killing CD133-positive cells. The results demonstrated that the CD133-targeted chemotherapy effectively reduced the viability of CD133-positive PA cells compared to non-targeted chemotherapy alone. In conclusion, the findings of this study provide strong evidence for the significant involvement of CD133 in chemotherapy resistance, not only in malignant brain tumors, as previously suggested, but also in low-grade gliomas, including pediatric PAs. 

These studies highlight the importance of targeting CD133 as a potential therapeutic strategy. The role of the CD133 protein as a delivery strategy for chemotherapeutic cargoes is described in the chapter dedicated to the nanotechnology-based delivery system, while the resensitizing of cancer stem cells to chemotherapy is detailed and presented in the following section.

### 3.3. Possibility to Enhance Immunotherapy

Tumor immunotherapy is an alternative modality of tumor treatment using tumor-specific antibodies and cellular immune effectors that aims to prevent tumor metastasis and improve individual quality of life. Tumor immunotherapy consists of passive immunotherapy, which depends on the repeated application of tumor antigen-specific antibodies or aptamers, and active immunotherapy, which relies on tumor-specific immune responses combining humoral and cytotoxic T cell effectors by the patient’s immune system [160]. The promising results of tumor immunotherapy have been reported in several studies.

Itai et al. established a novel anti-CD133 mouse monoclonal antibody of immunoglobulin G (IgG), namely CMab-43 (IgG2a, kappa), possessing a high sensitivity and specificity against immunized BALB/c nude mice with CD133-expressing LN229 brain glioblastoma cells [161]. Later, the anti-tumor activity of CMab-43 was investigated in colon carcinoma-derived cell lines (Caco-2) xenografted into female BALB/c nude mice. CMab-43 significantly lowered the volume and weight of the tumor in xenograft mice after day 12 compared with control IgG, while the body weight of mice was not altered. The results suggested that CMab-43 is beneficial for immunotherapy against CD133-expressing human colon cancers [162]. 

The study by Vora’s group showed that among three CD133-specific immunotherapies, including (i) IgG RW03, (ii) a dual-antigen T cell engager (DATE), and (iii) a second-generation chimeric antigen receptor T (CAR-T) cell (CAR133-T), CAR133-T demonstrated the most promising efficacy. CAR133-T cells were significantly proliferative after co-culture with CD133-expressing glioblastoma (GBM) cells, leading to an increase in tumor necrosis factor α (TNF-α) and interferon γ (IFN-γ) levels. In order to confirm the selective cytotoxicity of CAR133-T, cancer cell lysis was determined in CD133-expressing GBM cells. CAR133-T showed an increase in cancer cell lysis in a dose-dependent manner in two CD133-expressing GBM cells, GBM8 and BT935, while the control T cells did not show these effects. Moreover, the adverse effects of CAR133-T cells on normal CD133-expressing hematopoietic stem cells were not observed in humanized CD34-expressing NOD/SCID/IL2rγ^null^ (NSG) mice. No significant alterations of human CD45-, CD133-, or CD34-expressing hematopoietic engraftment were observed between the treatment of intracranial CAR133-T or control T cells [163].

Sangsuwannukul’s group developed a novel fourth-generation CAR133-T and tested its efficacy in high CD133-expressing human cholangiocarcinoma (CCA) cell line (KKU-213A cells). CAR133-T showed a potential cytotoxicity of up to 57.59 ± 9.62% cancer cell lysis in a dose-dependent manner in KKU-213A cells, whereas the treatment of untransduced T cells or CAR133-T in normal cholangiocytes did not. CAR133-T was confirmed to increase cytokine production, specifically in CD133-expressing CCA, via the significant upregulation of *IFNγ* and *TNFα* expression in KKU-213A cells compared to untransduced T cells. Hence, they concluded that CAR133-T is not only beneficial in CD133-expressing CCA but also for other CD133-expressing tumors. Notably, the KKU-100 cells, another high CD133-expressing CCA cell line with slightly lower CD133 expression than KKU-213A cells, did not significantly sensitize to CAR133-T [164].

Wang and coworkers showed the anti-tumor activities of CAR133-T cells using both in vitro, and in vivo pre-clinical and clinical studies. CAR133-T cells promoted cytotoxicity and production of IFN-γ and granzyme B after co-culture with tumor cell lines that highly expressed CD133 (i.e., SW1990, Hep3B, HT29, DLD1) without any effect towards non-expressing cells (i.e., SW480 and LOVO) when compared with co-culture with mock and untransduced T cells. CAR133-T cells significantly reduced the growth of HT29 cells with an increase in the copy of the CAR-T gene in the CD133-expressing xenograft BALB/c nude mice model compared with other groups. In the clinical study of an open-label and single-arm phase I trial, 3 and 14 out of 23 patients achieved partial remission (PR) and stable disease (SD), respectively, for 9 weeks to 15.7 months after treatment of CAR133-T cells in a dose of 0.5–2 × 10^6^/kg. Interestingly, a repeated cell infusion can extend a period of disease stability in PR and SD patients after the first infusion. The median progression-free survival (PFS) was 5 months. The 3- and 6-month disease control rates (DCRs) were 65.2 and 30.4%, respectively. Tumor remission was observed in nine patients, and de novo metastatic lesions were not detected in 21 patients during the trial. The liver biopsy showed the eradication of CD133-expressing tumor cells and the rapid proliferation of CD133-negative tumor cells after the cell infusion. It is important to note that almost every patient experienced hematologic toxicities, particularly thrombocytopenia, leukopenia, and hyperbilirubinemia (direct bilirubin) on days 3–5 after the cell infusion. Still, they can be self-recovered within 1 week [165]. 

In an open-label and single-arm phase II clinical trial, 1 and 14 out of 21 patients achieved PR and SD, respectively, after the first CAR133-T cell infusion. The median PFS was 6.8 months, and the overall survival (OS) was 12 months. The cytokine level was observed after the first cell infusion. TNF-α, interleukin (IL) 6, IFN-γ, vascular endothelial growth factor (VEGF), and stromal-cell-derived factor (SDF) 1 level were increased, whereas the level of a soluble form of VEGF receptors (VEGFR2) and platelet-derived growth factor (PDGF) BB, and endothelial progenitor cell (EPC) count were decreased. Notably, three and four patients suffered from Grade-2 thrombocytopenia and Grade-3 hyperbilirubinemia within 4 weeks after the first cell infusion [166]. 

Hu et al. explored a method of improving the efficacy of chimeric antigen receptor (CAR) T cells in an immunosuppressive tumor microenvironment by disrupting the programmed cell death protein 1 (PD-1) gene using CRISPR/Cas9-mediated genome editing [167]. The authors used nucleofection to deliver plasmids encoding both CRISPR/Cas9 for disrupting PD-1 and the piggyBac transposon system for expressing CD133-specific CAR in one reaction. The resulting PD-1-deficient CD133-specific CAR T cells showed improved proliferation, cytokine secretion, and cytotoxicity in vitro (U251 CD133-OE *luc* cells) and enhanced tumor growth inhibition in an orthotopic mouse model of glioma, compared to conventional CD133-CAR T cells. This method could be useful for producing PD-1-deficient CAR T cells for cancer immunotherapy. These trials showed effective and selective anti-tumor activity with the safety profile of CART-133 cells for patients with CD133-expressing tumors.

The Feng et al. study reports the case of a patient with advanced cholangiocarcinoma treated with a cocktail of chimeric antigen receptor (CAR)-modified T cells targeting both EGFR and CD133 [168]. The patient received three infusions of CAR T cells and showed a partial response, with a decrease in tumor size and a reduction in serum tumor markers. The treatment was well tolerated, and the patient experienced only mild adverse effects. The study suggests that a cocktail of CAR T cells targeting multiple antigens could be a promising approach for treating advanced cholangiocarcinoma.

### 3.4. Approach to Improve the Selectivity of Photodynamic Therapy

Photodynamic therapy (PDT) is a method of treatment that combines a non-toxic compound called photosensitizer (PS), oxygen, and light. None of these components have cytotoxic properties, but together they lead to generating reactive oxygen species (ROS) and/or free radicals [169]. Singlet oxygen is considered to be the main mediator starting the cascade of reactions leading to PDT-induced cell death. In the photodynamic therapy of CSCs, the goal is to destroy these cells to prevent the disease’s recurrence and increase the therapy’s effectiveness [170]. 

Yan et al. compared a photosensitizer pyropheophorbide-a (Pyro), which does not have tumor selectivity, with a novel photosensitizer, CD133–Pyro, obtained via the conjugation of Pyro to a peptide targeting CD133 [171]. The effectiveness of these PDT agents was tested in vitro on HT29 and SW620 cells and in vivo on SW620 cells. A new photosensitizer specifically targeted and enhanced the effectiveness of PDT in colorectal cancer stem cells. The study demonstrated that CD133-Pyro is more effective than unconjugated Pyro in inhibiting CD133^+^ CSCs. In animal testing, CD133-Pyro accumulated primarily in tumor tissue and had a high therapeutic efficacy. The mechanism of action of CD133-Pyro PDT involves the induction of ROS production, which ultimately leads to autophagic cell death. 

Photochemical internalization (PCI) is a promising approach that ensures efficient cellular transport and reaches the target side of molecules that do not readily penetrate the plasma membrane [172,173]. The PCI mechanism is based on the activation by light of photosensitizers sited in endocytic vesicles to release loaded macromolecules/drugs intracellularly [172]. This strategy can prevent lysosome degradation before the drug reaches the target site. Therefore, PCI uses photodynamic therapy to release the drug in the cytosol of cells. Olsen et al. used novel CD133-targeting immunotoxin (IT)—scFvCD133/rGelonin, in combination with endosomal escape method photochemical internalization (PCI) [174]. It was tested on different cell lines, including human colon adenocarcinomas (WiDr and HT29), human breast cancer (MDA-MB-231 and MCF-7), mouse embryonic fibroblast (NIH/3T3), and human glioblastoma (U87). The results show that scFvCD133/rGelonin effectively targets CD133 and induces cell death in CD133-expressing cancer cells. The authors suggest this approach could be a promising treatment strategy for CD133-expressing tumors.

Another study aimed to develop an effective PDT for cancer treatment using amino porphyrin–peptide assemblies [175]. It was found that these assemblies could effectively inhibit cancer stem cells. Moreover, the authors observed that amino porphyrin–peptide assemblies induced damage to ribosomes, the cell’s protein-making machinery, leading to cancer cell death. Interestingly, the study found that the expression of CD133, a marker for cancer stem cells, was reduced after treatment with amino porphyrin–peptide assemblies, indicating a potential role for this treatment in targeting cancer stem cells. The study suggests that amino porphyrin–peptide assemblies have significant potential for enhancing PDT as a cancer treatment option, particularly for targeting cancer stem cells.

### 3.5. Dealing with Cancer Stemness by Suppression CD133

The CD133 protein may be both a molecular target for the selective elimination of stem cells and a target in of itself. CD133 has been reported to be associated with chemoresistance in various cancer cells, including gastric [176], breast [177], colorectal [105], lung [178], ovarian [179], and glioma [180] cancer cells. Thus, targeting CD133 to sensitize the cells for chemotherapy or minimize tumor recurrence is a promising therapeutic strategy. It is even more significant since it was found that chemotherapeutic drugs can increase CD133 levels in cancer cells. Several in vitro studies showed that treatment with cisplatin [181,182], paclitaxel [183], and 5-fluorouracil (5-FU) [184,185,186] could be associated with the enlargement of the CD133-positive cell population. Liu et al. showed that the treatment of non-small cell lung cancer cell lines H460 and H661 with low-dose cisplatin enriched CD133-positive cells via NOTCH signaling [187]. Moreover, an increase in CD133 population upregulated ABCG2 and ABCB1 expression, which therefore increased the resistance to doxorubicin and paclitaxel [187]. Furthermore, in vivo, the study also confirmed that cisplatin treatment can increase the CD133-positive cells fraction, which can affect the therapy. The flow cytometry analysis of cells isolated from the xenografts showed a remarkable increase of 7 and 35 times in the CD133^+^ population in lung adenocarcinoma cell line A549 and CD133-negative adenocarcinoma lung LT66 tumors, respectively, seven days after chemotherapy with cisplatin [182]. Interestingly, in tumors derived from other cell lines LT45 and LT56, which are characterized by large populations of CD133 (50% and 15%, respectively), the number of CD133-positive cells was unchanged after cisplatin treatment, but a subpopulation of CD133^+^ABCG2^+^ cells—the potential chemoresistance clones—was increased. It is a crucial finding since that previous study also reported that patients with the dual expression of CD133 and ABCG2 are at higher risk for tumor recurrence [188].

In line with the drug repurposing strategy, metformin is one of the drugs that have gained lot of attention to affect CD133 expression in cancer cells. Several studies have indicated that metformin selectively affects CSCs by decreasing CD133^+^ cells [189,190,191]. Maehara et al. showed that metformin decreased the expression of CD133 via the AMPK/CCAAT enhancer-binding protein beta (CEBPβ) pathway. Using the HepG2 cell line, the authors found that metformin suppresses CD133 P1 promoter activity through upregulating the expression of CEBPβ, mainly the liver-enriched inhibitory protein (LIP) isoform [192]. The study conducted by Brown et al. aimed to evaluate the efficacy of metformin, a widely used anti-diabetic drug, as a cancer-stem-cell-targeting agent in ovarian cancer [193]. The researchers conducted a phase II clinical trial to assess the impact of metformin on ovarian cancer patients. They were assigned to receive either neoadjuvant metformin, debulking surgery, adjuvant chemotherapy and metformin, or neoadjuvant chemotherapy and metformin, interval debulking surgery, or adjuvant chemotherapy and metformin. The trial included patients with recurrent ovarian cancer who had previously received standard treatments. Metformin-treated tumors were evaluated for changes in CSC number and chemotherapy response compared to historical controls. The administration of metformin was well tolerated by the patients. The median progression-free survival was 18 months (95% CI 14.0–21.6), with a relapse-free survival at 18 months of 59.3% (95% CI 38.6–70.5). The median overall survival was 57.9 months (95% CI 28.0-not estimable). Tumors treated with metformin exhibited a 2.4-fold decrease in ALDH^+^CD133^+^ CSCs and increased sensitivity to cisplatin in ex vivo experiments. Additionally, metformin induced alterations in the methylation signature of cancer-associated mesenchymal stem cells (CA-MSCs), which prevented CA-MSC-driven chemoresistance in vitro. Also, the widely known antibiotic oxytetracycline can affect CD133 protein in cancer cells. Song and coworkers screened 3280 compounds selected from several libraries, such as the Library of Pharmacologically Active Compounds (LOPAC), and Prestwick and Enzo (FDA-approved compound) for a drug repositioning strategy [194]. The authors used immortalized hepatocyte line (Fa2N-4) and human liver cancer cells (Huh7.5) to identify the most cytotoxic and selective compounds. Based on these studies, the authors selected 13 compounds, while only four showed significant selective inhibition activity of both *α-fetoprotein* (AFP)^+^/CD133^+^ hepatocellular carcinoma (HCC) populations compared to the non-cancerous cell line [194]. The following compounds were selected for further studies (the drug target is presented in parenthesis): β-Chloro-L-alanine hydrochloride (alanine aminotransferase inhibitor), LY-294,002 (PI3K inhibitor), oxytetracycline (ribosomal protein synthesis inhibitor) and fusidic acid (GTPase-coupled protein synthesis inhibitor). A more detailed study showed that only one of the compounds mentioned above, oxytetracycline, could decrease the expression of CD133. The further experiments showed that oxytetracycline did not change the mRNA CD133. The experiment with protein synthesis inhibitor cycloheximide (CHX), which was used to evaluate the stability of the CD133 protein, showed that oxytetracycline might destabilize CD133 in the liver cancer stem cells.

Following a drug repurposing strategy, non-steroidal anti-inflammatory drugs (NSAIDs) have also been shown to be able to decrease CD133 levels. Moon and coworkers reported that indomethacin could modulate CD133 levels in colon-derived cancer cells [195]. Indomethacin treatment significantly decreased, with statistical significance, the CD133^+^CD44^+^ cell population in Caco-2 (7.0 to 4.8%) and SW620 (14.0 to 10.6%) [195]. On the other hand, treatment with 5-FU led to significant increases in CD133^+^CD44^+^ cells (Caco-2, 7.0 vs. 13.2%) and SW620 (14.0 to 25.6%) [195]. The combination of both drugs significantly reduced the proportion of CD133^+^CD44^+^ cells compared to treatment with 5-FU alone from 13.2 to 7.9%, and 25.6 to 17.7%, in Caco-2 and SW620 cells, respectively. The authors showed that the action of indomethacin is related to the downregulation of NOTCH/ hairy and the enhancer of split 1 (HES1) signaling pathway and also the upregulation of the expression of peroxisome proliferator-activated receptor γ (PPARG) [195]. 

It was also reported that acetylsalicylic acid might decrease the expression of ALDH1, Sox-2, Oct-4, CD44, and CD133 in human lung cancer cell lines and activate apoptosis and PTEN [196]. Deng et al. showed that CD133 expression was downregulated by celecoxib in two colon adenocarcinoma cell lines with different statuses of cyclooxygenase-2 (COX-2), DLD-1 (COX-2 negative) and HT29 (COX-2-positive) [197]. Mechanistic studies have shown that celecoxib may downregulate CD133 expression by affecting the Wnt pathway, which is associated with cancer stem cell differentiation [197]. The induction of differentiation is another promising approach to decreasing the CD133+ cells. De Carlo et al. showed that eicosapentaenoic acid (EPA) can decrease CD133 mRNA expression in colorectal adenocarcinoma (COLO 320 DM) cells [198]. The treatment with EPA resulted in both the downregulation of CD133 expression and upregulation of colonic epithelium differentiation markers cytokeratin 20 and mucin 2 [198]. These results confirmed that PUFA increased the differentiation status of colon cancer stem cells. The authors showed that EPA treatment could sensitize colon cancer cells to 5-fluorouracil treatment. The differentiating therapy can also be achieved using All-Trans Retinoic Acid (ATRA) [199]. It was reported that pretreatment with ATRA can reverse cisplatin resistance, specifically of the slowly dividing compartment, indicating an effect on CD133^+^/CXCR4^+^ cells in lung adenocarcinoma patient-derived xenograft model.

In their study, Song et al. discovered that the administration of chromenopyrimidinone (CPO) resulted in significant reductions in spheroid formation and the number of CD133^+^ cells in mixed HCC cell populations [200]. The effects of CPO were observed in HCC cells expressing varying levels of CD133, where it not only inhibited cell proliferation but also induced apoptosis and increased the expression of LC3-II. Additionally, CPO treatment led to point mutations in the ADRB1, APOB, EGR2, and UBE2C genes, resulting in the decreased expression of these proteins in HCC. Notably, among the four proteins, UBE2C expression was particularly controlled by CD133 expression in HCC. The researchers also injected Huh7 CD133^+^ cells into NOD/SCID mice. Despite its limited solubility, the administration of 5 mg/kg CPO effectively inhibited tumor growth without causing significant weight loss, as observed in mice treated with 10 mg/kg sorafenib. The study suggests that CPO could be a promising approach for treating hepatocellular carcinoma by CD133 suppression. The further study performed by the same group of Professor Seo obtained more insight about CPO-based therapy against CD133-overexpressing HCC cells in further study. The authors found that CD133 stabilized DNA methyltransferases (DNMT) activity via the regulation of DNA (cytosine-5)-methyltransferase 3 beta (DNMT3B) expression in several hepatocellular carcinoma cells [201].

Another one of the studies described involved downregulating CD133 expression in HepG2-CD133^+^ cells using lentivirus-mediated shRNA, followed by an analysis of the effects of CD133 on the modulation of stemness properties and chemoradiosensitivity in liver cancer stem cells (LCSCs) [202]. The findings of the study demonstrated that silencing CD133 in LCSCs significantly suppressed in vitro cell proliferation, tumorosphere formation, colony formation, and in vivo tumor growth in NOD/SCID mouse xenografts. Furthermore, the researchers observed that the suppression of CD133 increased the sensitivity of LCSCs to chemotherapy and radiotherapy. In conclusion, the study highlighted that targeting the stemness properties of LCSCs through CD133 presents a promising and novel strategy for the treatment of HCC. The results demonstrated that CD133 suppression not only hindered the proliferation and growth of LCSCs but also improved their responsiveness to chemotherapy and radiotherapy.

Li et al. investigated trilobatin anticancer efficacy in gefitinib-resistant lung cancer cells [203]. Trilobatin (phloretin-4-O-glucoside) is a dihydrochalcone glucoside and derivative of phloretin found in the stems, leaves, flowers and fruits of apple plants [204,205]. Trilobatin has been detected in the leaves of *Vitis species*, in *Lithocarpus polystachyus*, and in different *Malus* species including *Malus domestica* and *Malus trilobata*. Trilobatin is a strong natural sweetener and possesses pleitropic activity, such as anti-hyperglycemic [206], anti-inflammatory [207], anticancer [203], and antioxidant [208] properties. The results of the study demonstrated that trilobatin effectively inhibits the proliferation of these cells. Moreover, it increased the proportion of apoptotic cells and downregulated the expression levels of Bcl-2 and mitochondrial Cytochrome C while upregulating Bax, Cleaved Caspase-3, -9, and cytosolic Cytochrome C expression. Trilobatin also reduced tumor sphere formation and the expression levels of multiple stemness markers, including CD133.

## 4. Drug Delivery Systems for CD133-Targeted Therapy Based on Nanotechnology

Nanotechnology is a rapidly growing field that focuses on the design, characterization, production, and application of materials and devices at the nanoscale. It has potential applications in various fields, including medicine, materials science, and environmental science [209,210]. Nanoparticles have gained significant attention in cancer therapy because they selectively deliver therapeutic agents to cancer cells. Nanoparticles can be engineered to target specific cell types, including cancer stem cells expressing CD133. Several studies have explored the use of nanoparticles in CD133-targeted therapy [155,175,211,212,213,214,215,216,217,218]. The targeted delivery of drugs or genetic material through nanoparticles could increase the efficacy of CD133-targeted therapy, while minimizing off-target effects on healthy cells [210,219,220]. Nanoparticles can also help overcome drug resistance, a significant challenge in cancer treatment. In conclusion, the application of nanotechnology in CD133-targeted therapy holds significant promise for improving cancer treatment [220,221,222]. Further research is necessary to fully explore the potential of this approach in preclinical and clinical settings. The mechanisms of targeted delivery and possible delivery platforms are presented in Figure 5.

### 4.1. Gold Nanoparticles in CD133-Targeted Therapy

One of the most commonly used solutions in the CD133-targeted therapy of CSCs is the use of gold nanoparticles (AuNPs). Cho et al. reported developing a novel imaging agent using gold nanoparticles coated with peptide targeting the brain glioma stem cell marker CD133 [211]. The gold nanoparticles conjugated with a CBP4 peptide (specific to CD133) were tested on CD133-expressing glioma cells (U373) in vitro and in vivo. The results showed that the imaging agent could specifically target and label CD133-expressing cells and could potentially be used for the early detection and diagnosis of glioma. Additionally, the authors found that the imaging agent was biocompatible and did not cause toxicity or adverse effects in mice. Overall, the study provides promising evidence for the potential use of this imaging agent in diagnosing and treating glioma. 

The article of Poonaki et al. describes the development of a carrier platform for delivering clinical-stage GLS1 (glutaminase 1) inhibitor, Telaglenastat (CB-839), to glioblastoma stem cells [212]. The carrier platform consists of gold nanoparticles functionalized with the 15mer CD133 aptamer, generating Au-PEG-CD133-CB-839. The study showed that the CD133-functionalized gold nanoparticles could effectively deliver Telaglenastat to glioblastoma stem cells in vitro, significantly reducing cell viability and proliferation. The researchers concluded that this carrier platform could potentially improve the efficacy of Telaglenastat as a targeted therapy for cancer. 

Mohd-Zahid et al. also described a study where carboxyl-terminated PEGylated gold nanoparticles functionalized with anti-CD133 monoclonal antibody were used for selective and enhanced drug delivery [213]. AuNPs were loaded with the chemotherapeutic 5-fluorouracil to generate a targeted drug delivery system for colorectal cancer cells. An in vitro study on HCT116 cells suggested that functionalizing PEGylated AuNPs with antibodies and chemotherapeutic drugs facilitated cellular internalization and triggered cancer cell death and that anti-CD133 mAb conjugates have potential as a targeted drug delivery system for colorectal cancer treatment. The authors recommended optimizing the chemotherapeutic effects in future studies. 

Another study used gold nanoparticles conjugated with an antibody to deliver a phthalocyanine (aluminium (III) phthalocyanine chloride tetrasulphonate, AlPcS4Cl) for photodynamic therapy in lung cancer stem cells, which were irradiated by light with a wavelength of 673.2 nm [214]. The study results showed that the gold nanoparticle–anti-CD133 antibody conjugate could selectively target and deliver the photosensitizer drug to the cancer stem cells, resulting in a significant reduction in their viability compared to nanoparticles without antibody conjugates. The study also demonstrated that the PDT treatment led to a decrease in the expression of stem cell markers, indicating a potential therapeutic benefit in targeting lung cancer stem cells. 

Tan et al. used functionalized AuNPs (also called GNS@IR820/DTX-CD133) to treat castration-resistant prostate cancer (CRPC) using photothermal therapy, photodynamic therapy, and chemotherapy while monitoring it through near-infrared fluorescence and photoacoustic imaging [215]. The CD133 antibody helped deliver the drug specifically to the tumor tissues, enhancing the combined therapeutic effect. The nanoplatform showed antitumor effects in vitro and in vivo on the human prostate adenocarcinoma (PC3) cell line. The biodistribution of GNS@IR820/DTX-CD133 could be monitored with NIR imaging. As a multifunctional nanoplatform integrating different strategies with tumor imaging, GNS@IR820/DTX-CD133 has great potential for clinical use in CRPC therapy.

### 4.2. Nanoliposomes

Another way to specifically deliver therapeutics to CSCs is the use of nanoliposomes. Dadashi Noshahr et al. compared the effectiveness of Doxil (liposomal Doxorubicin) combined with anti-CD133 monoclonal antibodies using two different techniques for conjugation [216]. The optimized post-insertion method resulted in more antibodies conjugated per liposome than the routine post-insertion method, but both methods had similar drug release and leakage patterns. In vitro tests on the HT-29-CD133-positive cell line (colon cancer) showed that CD133-targeted Doxil had a significantly higher cellular uptake, binding, and internalization which lowered the inhibitory doxorubicin concentration compared to non-targeted Doxil. These results suggest that the specific recognition and binding of antibodies with CD133 receptors on cancer cells can enhance the efficacy of Doxil and provide a proof-of-principle for an active targeting concept. Further studies are needed to evaluate the in vivo efficacy of CD133-targeted Doxil. 

Also, in 2020, Wang’s group described a liposome-based delivery system, called CEP-LP@S/D, for the synergistic treatment of the human hepatocellular carcinoma (Huh-7) cell line [175]. The liposomes were dual-coated with CD133^-^ and EpCAM-targeted peptides, allowing for the selective targeting of CD133^+^ EpCAM^+^ liver CSCs. Once inside the CSCs, the liposome degraded via glutathione (GSH)-triggered disulfide bond breaking, releasing doxorubicin and salinomycin (Sal) to inhibit tumor growth through Dox-induced apoptosis and concurrent lysosomal iron sequestration by Sal. The system was found to effectively enhance CSC targeting, and eliminate the non-CSC fraction, exhibiting high antitumor efficacy in both in vitro and in vivo studies. The authors suggested that the smart liposome-based nanocarrier co-delivery system may be a promising strategy to combat liver cancer and other cancer types.

### 4.3. Other Delivery Systems

Other nanoparticle-based systems have also been used for CD133-targeted anti-cancer drug delivery. Ni et al. developed salinomycin-loaded polymeric nanoparticles conjugated with CD133 aptamers (Ap-Sal-NP) to specifically target and destroy CD133^+^ osteosarcoma (Saos-2) cancer stem cells [217]. The nanoparticles effectively killed CD133^+^ osteosarcoma CSCs both in vitro and in vivo. The authors confirmed that Ap-C6-NP specifically binds to CD133 antigen and is internalized into CD133-positive cells via receptor-mediated endocytosis. Furthermore, the mice xenograft study showed a 17.4-fold, 14.2-fold increase, and 7.1-fold increase in the tumor volume compared to the control group for Sal, Sal-NP, and Ap-Sal-NP, respectively [217]. The results suggested that Ap-Sal-NP has the potential to significantly inhibit osteosarcoma growth by killing CD133-expressing CSCs. In 2017, Huang et al. developed a targeted drug delivery system for gefitinib, a drug commonly used for treating lung cancer, to specifically target cancer stem cells [219]. They used DSPE-PEG2000 nanomicelles loaded with gefitinib and CD133 aptamers to target CD133. In vitro experiments on the A549 cell line demonstrated that the nanomicelles effectively targeted and killed lung CSCs, with enhanced cytotoxicity compared to free gefitinib. In vivo studies also showed that the nanomicelles significantly inhibited tumor growth and improved overall survival in a lung cancer mouse model. 

Wang et al. used carbon nanotubes (CNTs) conjugated with CD133 monoclonal antibody to target and destroy glioblastoma stem cells through photothermolysis [155]. The results showed that the CNT–CD133 conjugate is selectively bound to GSCs in vitro and in vivo and that photothermal therapy using near-infrared laser irradiation effectively killed the targeted GSCs while leaving the surrounding healthy cells intact. The study suggested that CNT–CD133 conjugates could be a promising therapeutic approach for treating glioblastoma using photothermal therapy.

These studies show promising evidence for the potential use of nanotechnology in the diagnosis and treatment of cancer (Table 2).

## 5. Limitations and Advantages

The ability to target cancer stem cells is still a significant challenge. However, searching for new possibilities and a better understanding of existing approaches might develop strategies to effectively eradicate CSCs, or prevent their potential to cause tumor recurrence. Using CD133 as a molecular target can offer an attractive therapeutic strategy to improve patient outcomes, and due to its extracellular location, it may be used as a target for drug delivery systems. On the other hand, CD133-targeted therapy has some limitations that must be considered to better design novel treatment regimens. 

One of the issues often discussed in the literature is the problem of the reliable detection of CD133 and the limitation of antibodies identified CD133-expressing cells [113,223]. Furthermore, it is widely known that CD133 expression depends on several factors, and small changes in the microenvironment can affect protein expression. The CD133 expression is modulated by oxygen level, cell density, or cell cycle phase [113,223,224]. Indeed, the detection of CD133 is mostly based on immunohistochemical methods and flow cytometry, which require antibodies for the accurate identification of CD133. As nicely described by Glumac and LeBeau [113], CD133 is highly sensitive to glycosylation modification, which may affect antibody binding. The most popular antibody clones used in CD133 detection, CD133/1 (AC133 or W6B3C1) and CD133/2 (AC141 or 293C3), bind to two different, glycosylated epitopes on the EC3 region of CD133 [113]. However, the alternative splicing and masking of the epitope-binding site via differential glycosylation might decrease detection accuracy [113]. Thus, to overcome the antibodies’ limitations in identifying CD133, the CD133-targeted aptamers can be used instead of antibodies. Ding and coworkers presented a novel “turn-on” FRET nano-aptamer sensor with CdSe/ZnS quantum dots (QDs) and gold nanoparticles as the energy donor–acceptor pairs with a detection limit of around 6.99 nM for CD133 detection [225]. The proposed sensor based on FRET occurred when the CD133 aptamer was hybridized with ssRNA, allowing one to bring QDs and AuNPs into proximity; then, the fluorescence of QDs was quenched by AuNPs. The fluorescence recovery of QDs was related to the ability of CD133 to competitively replace ssRNA and bind it to the CD133-targeted aptamer. Moreover, Zhang and coworkers designed a graphene–peptide-based fluorescent sensing system using a graphene oxide platform and a CD133-specific recognition peptide, with a linear range from 0 to 630 nM and a detection limit of 7.91 nM) [226]. Thus, designing a more specific detection system beyond the antibodies can overcome the limitation of the traditional method. However, further studies are needed to verify the reliability and effectiveness.

As it was reported for colon and glioblastoma cancer, both CD133-positive and CD133-negative cancer cells could initiate tumors [227,228]; thus, the role of CD133 as a marker for tumor-initiating stem cells is still ambiguous. However, despite the controversy regarding the usefulness of CD133 as a stem cell marker, numerous studies showed that higher levels of CD133 correlate with worse prognoses for patients. Joseph et al. showed that CD133 was significantly associated with poor prognostic characteristics in breast cancer patients, such as high histological grade, younger age, high Nottingham Prognostic Index, and estrogen and progesterone receptor negative subtypes that often are chemoresistant [177]. A similar observation was found for different cancers, such as gastric [28], colon [105], cervical [229], ovarian [108], and glioblastoma [230]. Therefore, CD133 as a molecular target is still an important “player” in targeted therapy. CD133-based therapy requires an understanding that this type of method is intended to complement another therapeutic approach. Due to the high heterogeneity of tumor tissue, only combination therapy that will allow for the elimination of rapidly proliferating and quiescent cells and can improve patient outcomes. Undoubtedly, implementing the principle of personalized medicine is necessary to move this therapy from bench to bedside. Since the targeted therapy is opposite to the still-realized model that one-size-fits-all; even this type of advanced treatment will not be successful without the genetic and molecular profiling of individuals.

## 6. Conclusions

In summary, targeting CD133 on CSCs is a promising therapeutic strategy for treating various types of cancer. Gene therapy, siRNA, and immunotherapy utilizing CD133-specific antibodies, aptamers, and CAR T cells have shown encouraging preclinical results. Nanotechnology-based therapies, such as AuNPs and nanoliposomes, can improve the efficacy of targeted therapies by delivering drugs specifically to CSCs (Table 2). However, further research is necessary to optimize the delivery of nanoparticles to CSCs, minimize off-target effects, and evaluate the safety of nanotechnology-based therapies. Moreover, one significant limitation of CD133-targeted therapy is the recycling of CD133 after transient silencing, necessitating additional investigations into the mechanisms of CD133 recycling and potential solutions to overcoming this limitation. On the other hand, it should be emphasized that CD133 can be used not only as a molecule that allows cargo delivery specifically to the CSC. Moreover, the depletion of CD133 by using a specific siRNA/shRNA, antibody, or chemotherapeutics has also been beneficial in cancer therapy. With continued research, CD133-targeted and nanotechnology-based therapies can potentially improve cancer treatment and enhance patient outcomes.

## Figures and Tables

**Figure 1 ijms-24-10910-f001:**
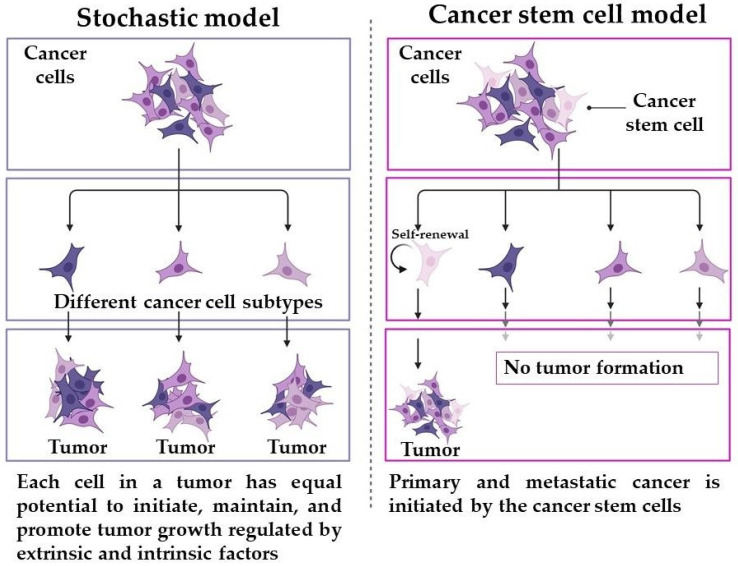
The theories of stochastic and CSC models in tumor development. Created with BioRender.com (accessed on 6 May 2023).

**Figure 2 ijms-24-10910-f002:**
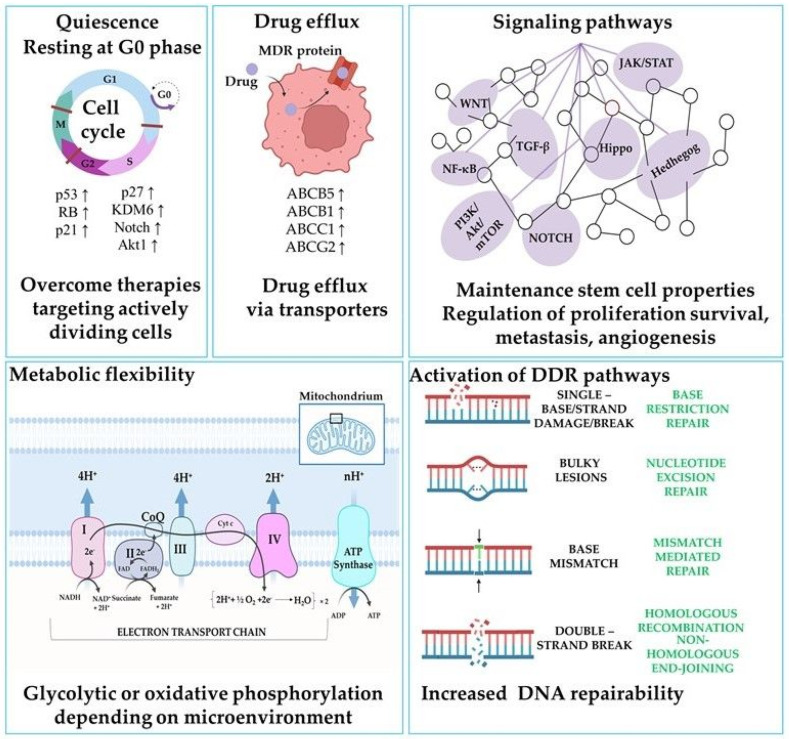
The mechanisms of resistance to chemo- and radiotherapy in cancer stem cells. Created with BioRender.com, (accessed on 6 May 2023). Abbreviations: ABCB5, ATP-binding cassette subfamily B member 5; ABCB1, ATP-binding cassette subfamily B member 1, ABCC1, ATP-binding cassette subfamily C member 1; ABCG2, ATP-binding cassette subfamily G member 2; CoQ, coenzyme Q; Cyt c, cytochrome c; DDR, DNA damage response; FAD, flavin adenine dinucleotide; KDM6, lysine demethylase 6; JAK/STAT, Janus kinase/signal transducer and activator of transcription; MDR, multidrug resistance; NAD, nicotinamide adenine dinucleotide; p21, cyclin-dependent kinase inhibitor 1; p27, cyclin-dependent kinase inhibitor 1B; RB, retinoblastoma protein; TGF-β, transforming growth factor beta.

**Figure 3 ijms-24-10910-f003:**
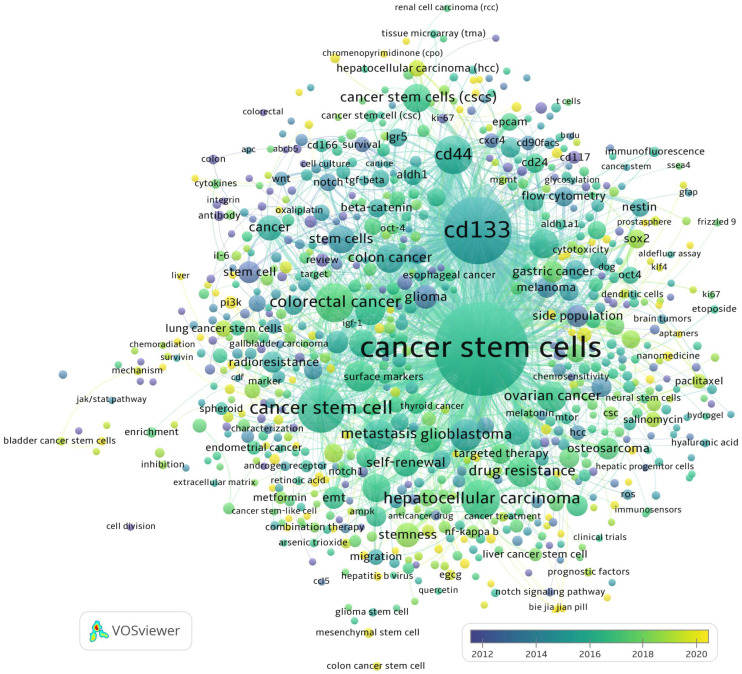
The co-occurrence of the key terms “cancer stem cell” and “CD133” in 3049 papers indexed in Web of Science published between 2004 and June 2023. The full records were exported from Web of Science as plain text and visualized using VOSviewer 1.6.19 https://www.vosviewer.com/ (accessed on 14 June 2023).

**Figure 4 ijms-24-10910-f004:**
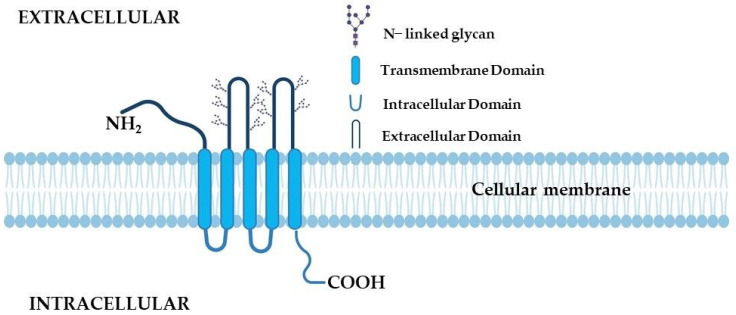
Schematic of the CD133 structure. Created with BioRender.com (accessed on 6 May 2023).

**Figure 5 ijms-24-10910-f005:**
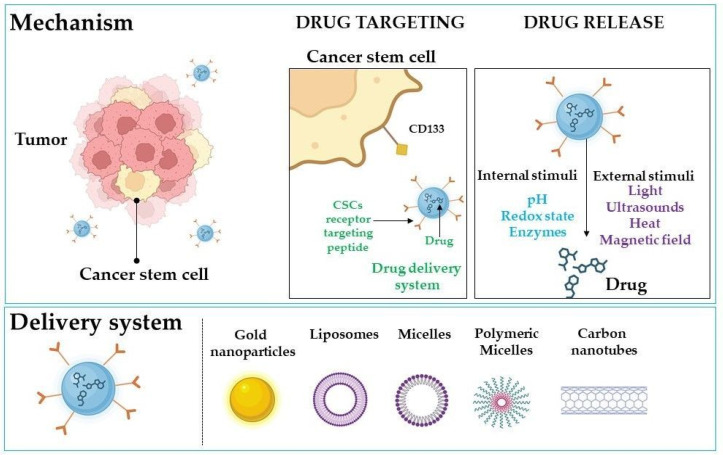
Selective targeting of CD133-expressing cancer stem cells using nanoparticles. Created with BioRender.com (accessed on 6 May 2023).

**Table 1 ijms-24-10910-t001:** Stem cell markers and their localization.

Cancer Type	Location	Marker	Refs.
Bladder	Extracellular/Surface	CD44, CD47, CEACAM-6	[20,21,22]
	Intracellular	ALDH1A1, SOX4	
Breast	Extracellular/Surface	CD44, CD24, CD133, CD49f, SSEA-3, CD70, PROCR, CD326, CD29, CD25, CD49, CD61, CD90E	[23,24]
	Intracellular	TSPAN8, ALDH1A1, BMI1, FOXO3, NANOG, NOTCH 1-3, OCT4, SOX2	
Colorectal	Extracellular/Surface	CD24, CD44, CD49, CD90, CD133, CD166, CD326, LGR5 CD29	[25,26,27]
	Intracellular	ALDH1A1, LETM1, NANOG, OCT4, SALL4, SOX2	
Gastric	Extracellular/Surface	CD24, CD44, CD90, CD133, CD326, LGR5, LINGO2	[28,29,30]
	Intracellular	ALDH1A1, NANOG OCT4, SOX2	
Glioblastoma	Extracellular/Surface	CD44, CD133, CD15, CD70	[31,32,33,34,35,36]
	Intracellular	S100A4, ALDH1A3, NANOG, OCT4, SOX2, Nestin, Musashi-1	
Liver	Extracellular/Surface	CD24, CD44, CD90, CD133, CD326	[37,38,39,40,41,42,43]
	Intracellular	AFP, NANOG, NOTCH 1-3, OCT4, SOX2	
Lung	Extracellular/Surface	CD44, CD87, CD133, CD166, CD326, CD117, CD90	[44,45,46,47]
	Intracellular	ALDH1A1, NANOG, OCT4	
Melanoma	Extracellular/Surface	CD20, CD133, CD166, CD279, ABCB5, ABCG2	[48,49,50]
Ovarian	Extracellular/Surface	CD24, CD44, CD117, CD133, CD326	[51,52,53,54,55]
	Intracellular	OCT4, NANOG, SOX2, ALDH1	
Pancreatic	Extracellular/Surface	CD24, CD44, CD133, CD326, ABCB1	[56,57]
	Intracellular	DCLK1, CXCR4, OCT4,	
Prostate	Extracellular/Surface	CD166, CD44, CD133, CD326, CD117, TACSTD2	[22,58,59]
	Intracellular	ALDH1A1, TGM2	
CML Leukemia	Extracellular/Surface	CD33, CD34, CD36, CD117, CD123, CD114 CD56, CD135 CD93	[60,61,62,63]
	Intracellular	ALDH	
AML Leukemia	Extracellular/Surface	CD33, CD34, CD123, CD244, CLL1, CD9, CD96, CD25, CD32	[64,65,66,67,68]
	Intracellular	ALDH1A1, NANOG, OCT4, SOX2	

Abbreviations: ABCB1, ATP-binding cassette subfamily B member 1; ABCB5, ATP-binding cassette subfamily B member 5; ABCG2, ATP binding cassette subfamily G member 2; AFP, alpha fetoprotein; ALDH1A1, aldehyde dehydrogenase 1 family, member A1; ALDH1A3, aldehyde dehydrogenase 1 family, member A3; AML, acute myeloid leukemia; BMI1, B cell-specific Moloney murine leukemia virus integration site 1; CLL1, C-type lectin-like molecule-1; CD9, cluster of differentiation antigen 9; CD117, cluster of differentiation 117; CD123, alpha chain of interleukin 3 receptor; CD133, cluster of differentiation 133; CD15, cluster of differentiation 15; CD166, cluster of differentiation 166; CD20, cluster of differentiation 20; CD224, cluster of differentiation 244; CD24, cluster of differentiation 24; CD25, cluster of differentiation 25; CD279, cluster of differentiation 279; CD29, cluster of differentiation 29; CD326, cluster of differentiation 326; CD34, cluster of differentiation 34; CD33, sialic-acid-binding Ig-like lectin 3; CD36, cluster of differentiation 36; CD44, cluster of differentiation 44; CD47, cluster of differentiation 47; CD49, cluster of differentiation 49; CD49f, cluster of differentiation 49f; CD61, cluster of differentiation 61; CD66c, cluster of differentiation 66c; CD70, cluster of differentiation 70; CD87, cluster of differentiation 87; CD90, cluster of differentiation 90; CD96, cluster of differentiation 96; CEACAM-6, carcinoembryonic antigen-related cell adhesion molecule 6; CML, chronic myeloid leukemia; CXCR4, C-X-C chemokine receptor type 4; DCLK1, doublecortin-like kinase 1; LETM1, leucine zipper and EF-hand-containing transmembrane protein 1; LGR5, leucine-rich repeat-containing G protein-coupled receptor 5; LINGO2, leucine-rich repeat and Ig domain-containing 2; NANOG, nanog homeobox; OCT-4, octamer-binding transcription factor 4; PLAUR, plasminogen activator, urokinase receptor; PROCR, protein C receptor; S100A4, S100 calcium-binding protein A4; SALL4, spalt-like transcription factor 4; SOX2, SRY-box transcription factor 2; SOX4, SRY-box transcription factor 4; SSEA-3, stage-specific embryonic antigen 3; TACSTD2, tumor-associated calcium signal transducer 2; TGM2, transglutaminase 2; TSPAN8, tetraspanin 8.

**Table 2 ijms-24-10910-t002:** Summarization of nanoparticle-based therapies.

Molecule Type	Therapy Type	Target	Model	Ref.
Gold nanoparticles	Imaging agent	CD133	In vitro and in vivo	[212]
Gold nanoparticles	Drug delivery	CD133	In vitro	[213]
Gold nanoparticles	Drug delivery	CD133	In vitro	[214]
Gold nanoparticles	Photodynamic therapy	Lung CSC	In vitro	[215]
Gold nanoparticles	Multimodal therapy	CRPC	In vitro and in vivo	[216]
Nanoliposomes	Drug delivery	CD133	In vitro	[217]
Liposomes	Drug delivery	CD133	In vitro	[175]
Polymeric nanoparticles	Drug delivery	CD133	In vitro	[218]
Nanomicelles	Drug delivery	CD133	In vitro and in vivo	[219]
Carbon nanotubes	Photothermal therapy	CD133	In vitro and in vivo	[155]

## Data Availability

Not applicable.

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
