# Peer review of "Unmasking the Deceptive Nature of Cancer Stem Cells: The Role of CD133 in Revealing Their Secrets"

_ijms, 2023, doi:10.3390/ijms241310910_

Round 1

Reviewer 1 Report

The present article, written by Pospieszna and colleagues, provides an extensive review on recent progresses on the role of CD133 in cancer stem cells general understanding and on novel therapeutic approach aiming to target this resilient population. The topics are well addressed producing a coherent argument about the subject and a focused description of the field.

In my opinion the “CD133 physiological functions - cell differentiation, proliferation, and survival” is underdeveloped. Maybe a new paragraph describing its interaction with other molecular pathways or other molecular signaling regulating CSC could be beneficial.

121 to 128 references needed. 198 to 203 references needed. 223 to 229 references needed. 308 to 327 references needed.

Please spell out the full term at its first mention, indicate its abbreviation in parenthesis and use the abbreviation from then on and be consistent in their use (e.g. Cancer stem cell).

Author Response

Reviewer 1

 We appreciate the time and effort that the Reviewers dedicated to providing feedback on our manuscript. We are grateful for the insightful comments and valuable improvements to our paper. We have incorporated the Reviewer’s suggestions, and those changes are highlighted in yellow in the manuscript.

Please see below, in blue, for a point-by-point response to the Reviewer’s comments and concerns.

The present article, written by Pospieszna and colleagues, provides an extensive review on recent progresses on the role of CD133 in cancer stem cells general understanding and on novel therapeutic approach aiming to target this resilient population. The topics are well addressed producing a coherent argument about the subject and a focused description of the field.

  1. In my opinion the “CD133 physiological functions - cell differentiation, proliferation, and survival” is underdeveloped. Maybe a new paragraph describing its interaction with other molecular pathways or other molecular signaling regulating CSC could be beneficial.

We thank the Reviewer for this valuable comment. Following the Reviewer's suggestion, we added new paragraph describing the relation of CD133 with CSC-related signaling pathways.

Lines 289-310  in the revised version:

“In addition to its role in maintaining stemness and self-renewal capacity in various stem cell populations, CD133 has been found to interact with several molecular pathways and signaling networks that regulate CSCs. One such pathway is the PI3K/Akt/mTOR pathway, which is crucial for cell growth, survival, and metabolism. CD133 has been shown to activate this pathway, promoting CSC proliferation and survival. This interaction also contributes to the resistance of CSCs to chemotherapy and targeted therapies [128]. Furthermore, CD133 has been implicated in regulating the epithelial-mesenchymal transition (EMT), a process associated with increased CSC properties, tumor invasion, and metastasis. CD133 expression has been linked to the induction of EMT-related transcription factors, such as Snail, Twist, and ZEB1, promoting CSC invasiveness and metastatic potential [129]. Additionally, CD133 has been shown to modulate the activity of the NF-κB pathway, a central regulator of inflammation and tumor progression. CD133 expression enhances NF-κB activation, increasing CSC survival and apoptosis resistance and promoting an inflammatory microenvironment that supports CSC maintenance [130]. Moreover, CD133 has been reported to interact with the TGF-β signaling pathway, which plays a pivotal role in CSC plasticity and immune evasion. CD133 expression contributes to TGF-β-induced CSC phenotypic changes and immunosuppressive effects, promoting tumor growth and immune resistance [131]. Collectively, these interactions highlight the intricate involvement of CD133 in multiple molecular pathways and signaling networks that regulate CSC properties and tumor progression, offering potential therapeutic targets for disrupting CSC-mediated tumor growth and therapy resistance.”

  1. 121 to 128 references needed. 198 to 203 references needed. 223 to 229 references needed. 308 to 327 references needed.

We thank the Reviewer for this comment and we apologize for those omissions. Indeed, several statement needed references, and missing citations have been added to the revised version.

“Interestingly, CSCs and embryonic stem cells share similar traits, particularly in their developmental signaling pathways, which control the self-renewal of stem cells [20]. Activation of these highly conserved pathways can is believed at least in part be responsible for their resistance [21]. CSCs rely on critical pathways such as Hedgehog, Wingless-related integration site (Wnt/β-catenin), Notch, Janus kinase/signal transducer and activator of transcription (JAK/STAT) and nuclear factor erythroid 2-related factor 2 (Nrf-2) and it is commonly accepted that their dysregulation leads to differences in treatment resistance, metastasis, and proliferation between cancerous and normal stem cells [22–25].”

“CD133 has been shown to play a critical role in maintaining of stemness and self-renewal capacity in embryonic and neural stem cells [112,113]. CD133 expression has been observed in several tissues during embryonic growth, including the developing nervous system, retina, kidney, and liver [114].”

“Other antibodies, such as the anti-sialyl-di-Lewis- antibody FG129, are being tested to target tumor-associated glycans and develop tumor-selective treatments and diagnostic modalities [141]. Additional research is needed to distinguish the glycome and glycosylation differences among normal cells, stem cells, CSCs, and non-CSCs, despite recognizing of glycosylation's crucial involvement in CSC signalling pathways and markers regulating self-renewal, stemness, and extravasation. This information may enable the development of biomarkers for detecting cancer progression and allow researchers to accurately target cancer cells and resistant CSCs. Increasing evidence shows that abnormal glycosylation of CSCs plays a critical role in their ability to resist chemotherapy and metastasize through several pathways. Although inhibiting or manipulating glycosylation in CSCs has demonstrated therapeutic potential, further exploration of the associated glycosylation processes is necessary to devise effective strategies targeting specific altered markers or signalling pathways without affecting healthy cells [138]. Approaches such as selectively cleaving the surface glycan of tumors or drugs with an affinity for tumor-associated glycans have demonstrated varying toxicity to cancer and normal cells, indicating potential therapeutic window optimization [142,143]. Although CSC markers may exhibit intratumoral and intertumoral heterogeneities, glycosylation may provide relevant targets, such as sialyl acid, pre-served throughout tumors, simplifying the development of effective and extensive treatment strategies [138].”

  1. Please spell out the full term at its first mention, indicate its abbreviation in parenthesis and use the abbreviation from then on and be consistent in their use (e.g. Cancer stem cell).

We thank the Reviewer for this notice and apologize for this oversight. The manuscript has been reviewed to introduce all abbreviations in the right places.

Reviewer 2 Report

A better organization of the paper should bhelp in the reading of the text.

It is not clear is all the therapy based on CD133 targing is presented in this review. This woudl help in understading the relevance and usefulnes sof the work. At the moment it seems only partially usefull, non indication on the scientific relevance (statistics methods) are provided for the cited paper.

A general vision of the authors in a larger discussion could help, what is lacking is the take home message of this paper, what really need for CD133 targeted therapy? Which are the current limitations? How to deal with them?

English is fluent, only limited part of the text need revision as sentences seem to long.

Author Response

Reviewer 2

  1. A better organization of the paper should help in the reading of the text.

We thank the Reviewer for this suggestion. We have tried to add a short introduction that simultaneously explains the purpose or relevance of discussing a presented problem. We changed the titles of some chapters and section:

  1. A brief 26-year history of CD133: from discovery to understanding the role of protein.

(in the previous version: The discovery of CD133).

  1. Drug delivery systems for CD133-targeted therapy based on nanotechnology

(in the previous version: Nanoparticles in CD133-targeted therapy).

2.4 The role of CD133 glycosylation.

(in the previous version 3.2 Glycans as a target); moreover, we moved this section to chapter 2.

We also added new sections:

3.2 The relationship between CD133 and chemotherapeutic drugs (lines 407-444 in the revised version);

3.5 Dealing with cancer stemness by suppression CD133 (lines 577-717 in the revised version).

We hope that those modifications will improve our paper and make it easier for the readers to understand the main messages of this paper.

  1. It is not clear is all the therapy based on CD133 targeting is presented in this review. This woudl help in understading the relevance and usefulnes sof the work. At the moment it seems only partially usefull, non indication on the scientific relevance (statistics methods) are provided for the cited paper.

We thank Reviewer for this comment. According to the Reviewer’s suggestion, we added a new paragraph describing the CD133-based therapy. We also supposed that the notice regarding the statistic method could relate to the method of literature search and preparation PRISMA flow diagram to report on systematic review. Instead of this type of presentation, we prepare Figure 3 using VOSviewer 1.6.19:

Lines 149-158 in the revised version:

“The fundamental trends in the co-occurrence of key terms "cancer stem cells" and "CD133" with other key terms in the papers published in the last 10 years are shown in Figure 3. “

Figure 3.  The cooccurrence of the key terms “cancer stem cell” and “CD133” in 3049 papers indexed in Web of Science published between 2004 and June 2023. The full records were exported from Web of Science as plain text and visualized using VOSviewer 1.6.19 https://www.vosviewer.com/.

Lines 408-444 in the revised version:

“Cancer chemotherapy is a widely used treatment modality in the management of cancer. It involves the administration of specific drugs that have the ability to kill or inhibit the growth of cancer cells. Chemotherapy aims to target cancer cells through-out the body, including those that may have spread from the primary tumor to other sites, to achieve a therapeutic response and improve patient outcomes [148]. Several studies have shown that directly targeting CD133 with monoclonal antibody, aptamer, antibody fragments, and other advanced delivery system might transport chemotherapeutic drugs effectively to cancer stem cells [149–153].

The study conducted by Zhou et al. investigated the effects of CD133 expression on chemotherapy response and drug sensitivity in adenoid cystic carcinoma (ACC) [154]. The study aimed to explore the role of CD133 in determining the efficacy of chemotherapy in ACC. CD133 expression levels were analyzed in ACC tumor samples and correlated them with the response to chemotherapy and drug sensitivity. It was found that high CD133 expression was significantly associated with reduced response to chemotherapy and increased resistance to drugs commonly used in ACC treatment [154]. The study highlights the potential significance of CD133 as a predictive biomarker for chemotherapy response in ACC. The findings suggest that targeting CD133-positive cancer stem cells may improve the effectiveness of chemotherapy and overcome drug resistance in ACC patients.

The other study aimed to investigate the impact of targeting CD133 on the chemo-therapeutic efficacy of recurrent pediatric pilocytic astrocytoma (PA) following pro-longed chemotherapy [155]. The researchers sought to determine whether CD133 could be a potential therapeutic target to enhance treatment outcomes in this brain tumor. The study utilized in vitro models to evaluate the effects of targeting CD133 in recurrent PA. Firstly, the presence of CD133-positive cells in patient-derived PA samples was confirmed. They then examined the efficacy of CD133-targeted chemotherapy (doxorubicin, vinblastine, vincristine) in killing CD133-positive cells. The results demonstrated that the CD133-targeted chemotherapy effectively reduced the viability of CD133-positive PA cells compared to non-targeted or chemotherapy alone. In conclusion, the findings of this study provide strong evidence for the significant involvement of CD133 in chemotherapy resistance, not only in malignant brain tumors, as previously suggested, but also in low-grade gliomas, including pediatric PAs.

These studies highlight the importance of targeting CD133 as a potential therapeutic strategy. The role of the CD133 protein as a delivery strategy for chemotherapeutic cargoes is described in the chapter dedicated to the nanotechnology-based de-livery system, while the resensitizing of cancer stem cells to chemotherapy is detailed and presented in the below section.”

Lines 577-717 in the revised version:

“3.5 Dealing with cancer stemness by suppression CD133

The CD133 protein may be both a molecular target for the selective elimination of stem cells and a target in itself. CD133 has been reported to be associated with chemo-resistance in various cancer cells, including gastric [172], breast [173], colorectal [57], lung [174], ovarian [175], glioma [176] cancer cells. Thus, targeting CD133 to sensitize the cells for chemotherapy or minimizing the tumor recurrence is a promising therapeutic strategy. It is even more significant since it was found that chemotherapeutic drugs can increase CD133 levels in cancer cells. Several in vitro studies showed that treatment with cisplatin [177,178], paclitaxel [179], and 5-fluorouracil (5-FU) [180–182] could be associated with the enlargement of the CD133-positive cell population. Liu et al. showed that treatment of non–small cell lung cancer cell lines H460 and H661 with low-dose cisplatin enriched CD133-positive cells via NOTCH signaling [183]. Moreover, increase CD133 population upregulated ABCG2 and ABCB1 expression, which therefore increased the resistance to doxorubicin and paclitaxel [183]. Furthermore, in vivo, study also confirmed that cisplatin treatment can increase the CD133-positive cells fraction, which can affect the therapy. The flow cytometry analysis of cells isolated from the xenografts showed a remarkable increase of 7 and 35 times in the CD133+ population in lung adenocarcinoma cell line A549 and CD133-negative adenocarcinoma lung LT66 tumors, respectively, seven days after chemotherapy with cisplatin [178]. Interestingly, in tumors derived from another cell lines LT45 and LT56 which are characterized by large populations of CD133 (50% and 15%, respectively), the number of CD133-positive cells were unchanged after cisplatin treatment, but a subpopulation of CD133+ABCG2+ cells – the potential chemoresistance clones was in-creased. It is a crucial finding since that previous study also reported that patients with the dual expression of CD133 and ABCG2 are at higher risk for tumor recurrence [184].

In line with the drug repurposing strategy, metformin is one of the drugs that have gained lot of attention to affect CD133 expression in cancer cells. Several studies have indicated that metformin selectively affects CSCs by decreasing CD133+ cells [185–187]. Maehara et al. showed that metformin decreased the expression of CD133 via the AMPK/ CCAAT enhancer binding protein beta (CEBPβ) pathway. Using the HepG2 cell line, the Authors found that metformin suppresses CD133 P1 promoter activity through upregulating the expression of CEBPβ, mainly the liver-enriched inhibitory protein (LIP) isoform [188]. The study conducted by Brown et al. aimed to evaluate the efficacy of metformin, a widely used anti-diabetic drug, as a cancer stem cell-targeting agent in ovarian cancer [189]. The researchers conducted a phase II clinical trial to assess the impact of metformin on ovarian cancer patients. They were assigned to receive either neoadjuvant metformin, debulking surgery, adjuvant chemotherapy plus metformin, or neoadjuvant chemotherapy and metformin, interval debulking surgery, adjuvant chemotherapy plus metformin. The trial included patients with recur-rent ovarian cancer who had previously received standard treatments. Metformin-treated tumors for changes in CSC number and chemotherapy response compared to historical controls were evaluated. The administration of metformin was well tolerated by the patients. The median progression-free survival was 18 months (95% CI 14.0-21.6), with a relapse-free survival at 18 months of 59.3% (95% CI 38.6-70.5). The median overall survival was 57.9 months (95% CI 28.0-not estimable). Tumors treated with metformin exhibited a 2.4-fold decrease in ALDH+CD133+ CSCs and increased sensitivity to cisplatin in ex vivo experiments. Additionally, metformin induced alterations in the methylation signature of cancer-associated mesenchymal stem cells (CA-MSCs), which prevented CA-MSC-driven chemoresistance in vitro. Also, the widely known antibiotic oxytetracycline can affect CD133 protein in cancer cells. Song et co-workers screened 3,280 compounds selected from several libraries, such as Library of Pharmacologically Active Compounds (LOPAC), Prestwick and Enzo (FDA-approved com-pound) for drug repositioning strategy [190]. The authors used immortalized hepatocyte line (Fa2N-4) and human liver cancer cells (Huh7.5) to identify the most cytotoxic and selective compounds. Based on these studies, authors selected 13 compounds, while only four showed significant selective inhibition activity of the both α-fetoprotein (AFP)+/CD133+ hepatocellular carcinoma (HCC) population compared to the non-cancerous cell line [190]. The following compounds were selected for further studies (drug target is presented in parenthesis): β-Chloro-L-alanine hydrochloride (alanine aminotransferase inhibitor), LY-294,002 (PI3K inhibitor), oxytetracycline (ribosomal protein synthesis inhibitor) and fusidic acid (protein synthesis inhibitor GTPase coupled). A more detailed study showed that only one of the compounds mentioned above, oxytetracycline, could decrease the expression of CD133. The further experiments showed that oxytetracycline did not change the mRNA CD133. The experiment with protein synthesis inhibitor cycloheximide (CHX) which was used to evalu-ate the stability of CD133 protein showed that oxytetracycline might destabilize CD133 in the liver cancer stem cells.

Following a drug repurposing strategy, non-steroidal anti-inflammatory drugs (NSAIDs) have also been shown to be able to decrease CD133 levels. Moon and co-workers reported that indomethacin could modulate CD133 levels in colon-derived cancer cells [191]. Indomethacin treatment significantly decreased with statistical significance CD133+CD44+ cells population in Caco-2 (7.0 to 4.8%) and SW620 (14.0 to 10.6%) [191]. On the other hand, treatment with 5-FU led to significant increases in CD133+CD44+ cells (Caco-2, 7.0 vs. 13.2%) and SW620 (14.0 to 25.6%) [191]. The com-bination of both drugs significantly reduced the proportion of CD133+CD44+ cells com-pared to treatment with 5-FU alone from 13.2 to 7.9%, and 25.6 to 17.7%, in Caco-2 and SW620 cells, respectively. The authors showed that the action of indomethacin is related to downregulation of NOTCH/ hairy and enhancer of split 1 (HES1) signaling pathway and upregulation the expression of peroxisome proliferator-activated receptor γ (PPARG) [191].

It was also reported that acetylsalicylic acid might decrease the expression of ALDH1, Sox-2, Oct-4, CD44, and CD133 in human lung cancer cell lines and activate apoptosis and PTEN [192]. Deng et al. showed that CD133 expression was downregulated by celecoxib in two colon adenocarcinoma cell lines with different status of cy-clooxygenase-2 (COX-2) , DLD-1 (COX-2 negative) and HT29 (COX-2-positive) [193]. Mechanistic studies has shown that celecoxib may downregulate CD133 expression by affecting the Wnt pathway, which is associated with cancer stem cell differentiation [193]. The induction of differentiation is another promising approach to decreasing the CD133+ cells. De Carlo et al. showed that eicosapentaenoic acid (EPA) can decrease CD133 mRNA expression in colorectal adenocarcinoma (COLO 320 DM) cells [194]. The treatment with EPA resulted in both the down-regulation of CD133 expression and up-regulation of colonic epithelium differentiation markers cytokeratin 20 and mucin 2 [194]. These results confirmed that PUFA increased the differentiation status of colon cancer stem cells. The authors showed that EPA treatment could sensitize colon cancer cells to 5-fluorouracil treatment. The differentiating therapy can also be achieved using All-Trans Retinoic Acid (ATRA) [195]. It was reported that pretreatment with ATRA can reverse cisplatin resistance, specifically of the slowly dividing compartment, indicating an effect on CD133+/CXCR4+ cells in lung adenocarcinoma patient-derived xenograft model.

In their study, Song et al. discovered that the administration of chromenopyrimidinone (CPO) resulted in significant reductions in spheroid formation and the number of CD133+ cells in mixed HCC cell populations [196]. The effects of CPO were observed in HCC cells expressing varying levels of CD133, where it not only inhibited cell proliferation but also induced apoptosis and increased the expression of LC3-II. Additionally, CPO treatment led to point mutations in the ADRB1, APOB, EGR2, and UBE2C genes, resulting in decreased expression of these proteins in HCC. Notably, among the four proteins, UBE2C expression was particularly controlled by CD133 expression in HCC. The researchers also injected Huh7 CD133+ cells into NOD/SCID mice. Despite its limited solubility, the administration of 5 mg/kg CPO effectively inhibited tumor growth without causing significant weight loss, as observed in mice treated with 10 mg/kg sorafenib. The study suggests that CPO could be a promising approach for treating hepatocellular carcinoma by CD133 suppression. The further study performed by the same group of professor Seo got more insight about CPO-based therapy against CD133-overexpressing HCC cells in further study. The authors found that CD133 stabilized DNA methyltransferases (DNMT) activity via regulation of DNA (cytosine-5)-methyltransferase 3 beta (DNMT3B) expression in several hepatocellular carcinoma cells [197].

Another study described involved downregulating CD133 expression in HepG2-CD133+ cells using lentivirus-mediated shRNA, followed by an analysis of the effects of CD133 on the modulation of stemness properties and chemoradiosensitivity in liver cancer stem cells (LCSCs) [198]. The findings of the study demonstrated that silencing CD133 in LCSCs significantly suppressed in vitro cell proliferation, tumorosphere formation, colony formation, and in vivo tumor growth in NOD/SCID mouse xenografts. Furthermore, the researchers observed that the suppression of CD133 increased the sensitivity of LCSCs to chemotherapy and radiotherapy. In conclusion, the study high-lighted that targeting the stemness properties of LCSCs through CD133 presents a promising and novel strategy for the treatment of HCC. The results demonstrated that CD133 suppression not only hindered the proliferation and growth of LCSCs but also improved their responsiveness to chemotherapy and radiotherapy.

Li et al. investigated trilobatin anticancer efficacy in gefitinib-resistant lung can-cer cells [199]. Trilobatin (phloretin-4-O-glucoside) is a dihydrochalcone glucoside and derivative of phloretin found in the stems, leaves, flowers and fruits of apple plants [200,201]. Trilobatin has been detected in the leaves of Vitis species, in Lithocarpus polystachyus, and in different Malus species including Malus domestica and Malus trilobata. Trilobatin is a strong natural sweetener possesses pleitropic activity, such as anti-hyperglycemic [202], anti-inflammatory [203], anticancer [204], and antioxidant [205] properties. The results of the study demonstrated that trilobatin effectively inhibits the proliferation of these cells. Moreover, it increased the proportion of apoptotic cells and down-regulated the expression levels of Bcl-2 and mitochondrial Cytochrome C while up-regulating Bax, Cleaved Caspase-3, -9, and cytosolic Cytochrome C expression. Trilobatin also reduced tumor sphere formation and the expression levels of multiple stemness markers, including CD133.”

  1. A general vision of the authors in a larger discussion could help, what is lacking is the take home message of this paper, what really need for CD133 targeted therapy? Which are the current limitations? How to deal with them?

We agree with the Reviewer that the review should be modified to get a clear picture about the CD133-targeted therapy. According to the suggestions the following paragraphs were added to the manuscript:

Lines 159-170 in the revised version:

“The above figure shows how much the CD133 protein has been studied in different aspects. These studies aimed better to understand its properties and biological functions concerning cancer disease. All this research gives a huge step to transferring the knowledge to practical application and might open new ways for cancer treatment and overcoming the resistance to currently used therapies. In this review, we briefly described the functions of CD133 and its role in physiological processes and the “dark side” in cancer development and resistance to treatment. We discussed the possibilities of using the CD133 as a molecular target and focused on gene therapy, immunotherapy, chemotherapy, and photodynamic therapy. These treatment options were selected as the most studied in recent years and seemed more promising to improve therapeutic effectiveness. Finally, we focused on the CD133 as a drug delivery system based on nanocarriers for different cargoes to target cancer stem cells selectively.”

Lines 362-366 in the revised version:

“To date, by targeting CD133 in cancer cells, different cargoes can be delivered specifically to the cancer stem cell population. On the other hand, affecting the CD133 gene expression using the small interfering RNA (siRNA)/short hairpin RNA (shRNA) or chemotherapeutics can reduce the stemness potential of cancer cells. “

Lines 861-882 in the revised version:

“Indeed, the detection of CD133 is mostly based on immunohistochemical methods and flow cytometry, which require antibodies for accurate identification of CD133. As nicely described by Glumac and LeBeau [113], CD133 is highly sensitive to glycosylation modification, which may affect antibody binding. The most popular antibody clones used in CD133 detection: CD133/1 (AC133 or W6B3C1) and CD133/2 (AC141 or 293C3), bind to two different, glycosylated epitopes on the EC3 region of CD133 [113]. However, the alternative splicing and masking epitope binding site via differential glycosylation might decrease the detection accuracy [113]. Thus, to overcome the antibodies' limitations in identifying CD133, the CD133-targeted aptamers can be used instead of antibodies. Ding and co-workers presented a novel "turn-on" FRET nano-aptamer sensor with CdSe/ZnS quantum dots (QDs) and gold nanoparticles as the energy donor -acceptor pairs with a detection limit of around 6.99 nM for CD133 detection [223]. The proposed sensor based on FRET occurred when the CD133 aptamer was hybridized with ssRNA, allowing one to bring QDs and AuNPs into proximity; then, the fluorescence of QDs was quenched by AuNPs. The fluorescence recovery of QDs was related to the ability of CD133 to competitively replace ssRNA and bind it to the CD133-targeted aptamer. Moreover, Zhang and co-workers designed a graphene-peptide-based fluorescent sensing system using a graphene oxide platform and a CD133-specific recognition peptide, with a linear range from 0 to 630 nM and a detection limit of 7.91 nM) [224]. Thus, designing a more specific detection system beyond the antibodies can overcome the limitation of the traditional method. However, further studies are needed to verify the reliability and effectiveness.”

Lines 894-901 in the revised version:

“CD133-based therapy requires an understanding that this type of method is intended to complement another therapeutic approach. Due to the high heterogeneity of tumor tissue, only combination therapy that will allow the elimination of rapidly proliferating and quiescent cells can improve patient outcomes. Undoubtedly, implementing the principle of personalized medicine is necessary to move this therapy from bench to bedside. Since the targeted therapy is opposite to still realize model that one-size-fits-all, even this type of advanced treatment will not be successful without genetic and molecular profiling of individuals.”
